# Development and Evaluation of a Duplex Lateral Flow Assay for the Detection and Differentiation between Rabbit Haemorrhagic Disease Virus *Lagovirus europaeus*/GI.1 and /GI.2

**DOI:** 10.3390/biology11030401

**Published:** 2022-03-05

**Authors:** Alba Fresco-Taboada, Mercedes Montón, Istar Tapia, Elena Soria, Juan Bárcena, Cécile Guillou-Cloarec, Ghislaine Le Gall-Reculé, Esther Blanco, Paloma Rueda

**Affiliations:** 1Eurofins Ingenasa S.A., 28037 Madrid, Spain; mmonton@ingenasa.com (M.M.); itapia@ingenasa.com (I.T.); esoria@ingenasa.com (E.S.); prueda@ingenasa.com (P.R.); 2Centro de Investigación en Sanidad Animal (CISA-INIA/CSIC), Valdeolmos, 28130 Madrid, Spain; barcena@inia.es (J.B.); blanco@inia.es (E.B.); 3French Agency for Food, Environmental and Occupational Health & Safety (Anses), Avian and Rabbit Virology, Immunology and Parasitology Unit, Ploufragan-Plouzané-Niort Laboratory, 22440 Ploufragan, France; cecile.guillou-cloarec@anses.fr (C.G.-C.); ghislaine.legall-recule@anses.fr (G.L.G.-R.)

**Keywords:** Rabbit haemorrhagic disease virus, lateral flow assay, antigen detection

## Abstract

**Simple Summary:**

Rabbit Haemorrhagic Disease is caused by a virus that affects the liver, the spleen and the lungs of rabbits, causing hepatitis, splenomegaly and haemorrhages. A new genotype of the virus was first reported in France in 2010 and has spread globally since then, replacing most of the circulating former viruses in many countries. The detection of the virus and the differentiation of both genotypes is of crucial importance for disease surveillance. In this article, a rapid test for antigen detection is described and evaluated, providing the first description of a quick and easy-to-use test that allows for the simultaneous detection and differentiation of the genotypes. A total of 136 samples, rabbit liver samples and liver exudates (liquid collected after freeze–thawing) classified as infected and non-infected, were analysed, with good results. These data confirm that the developed rapid test can be used as a reliable diagnostic test for disease surveillance, especially in farms and the field.

**Abstract:**

Rabbit Haemorrhagic Disease Virus 2 (RHDV2, recently named *Lagovirus europaeus*/GI.2) was first reported in France in 2010 and has spread globally since then, replacing most of the circulating former RHDV (genotype GI.1) in many countries. The detection and differentiation of both genotypes is of crucial importance for the surveillance of the disease. In this article, a duplex lateral flow assay (LFA) for antigen detection is described and evaluated, providing the first description of a quick and easy-to-use test that allows for the simultaneous detection and differentiation of RHDV genotypes GI.1 and GI.2. A panel of GI.1- or GI.2-infected and non-infected rabbit liver samples and liver exudates (136 samples) was analysed, obtaining a total sensitivity of 94.4% and specificity of 100%. These data confirm that the developed duplex LFA can be used as a reliable diagnostic test for RHD surveillance, especially in farms and the field.

## 1. Introduction

Rabbit Haemorrhagic Disease Virus (RHDV), a member of the genus *Lagovirus* (family *Caliciviridae*), causes a fatal disease in domestic and wild European rabbits (*Oryctolagus cuniculus*) and was first reported in China in 1984. Since then, it has been reported in over forty countries in the five continents, being endemic in most parts of the world where European rabbits are domesticated. From 70% to 90% of infected rabbits over eight weeks of age die within 12–36 h. The primary target organs of the virus are the liver, the spleen and the lungs. RHD is characterized by acute necrotizing hepatitis, splenomegaly and haemorrhages in several organs, and is generally associated with massive disseminated intravascular coagulation (reviewed in [1,2]). In 2010, a new pathogenic lagovirus, phylogenetically distinct from any previously described pathogenic and benign members of the genus, was identified in France [3]. It presents with a unique antigenic profile [4] and was recently classified as a second RHDV genotype [2]. Since then, it has rapidly spread throughout other European countries, then Australia, Africa, and North America (reviewed in [5]), replacing most of the circulating RHDV in the majority of these countries [4,5,6,7,8]. It was first reported in Asia in April 2020 [9]. This new genotype was first designated as RHDV2 or RHDVb, according to the authors. A recent proposal of a unified nomenclature for lagoviruses defined a single species of lagovirus (*Lagovirus europaeus*) and classified RHDV and RHDV2 within genogroup I as genotype 1 (GI.1) and genotype 2 (GI.2), respectively [10]. GI.2 causes a clinical disease, similar to that caused by GI.1 [11]. However, GI.2 differs from GI.1, affecting a wide range of hare and jackrabbit species (*Lepus* spp.), several cottontail rabbit species (*Sylvilagus* spp.) and other native American lagomorphs, in which it produces similar lesions [11,12,13,14,15,16,17,18]. It has also been identified in some non-lagomorphs [19,20]. In addition, GI.2 can affect rabbits as young as 11 days (pre-weaned) [21], as opposed to GI.1, which is infrequently seen in rabbits under 8–10 weeks of age. The first results indicated that early GI.2 induced a higher prevalence of subacute and chronic forms than GI.1, as well as a longer disease. It was also stated that, unlike GI.1, which can cause death in most rabbits, a variable rate of mortality (from 5 to 70%) was observed in rabbits infected with GI.2 [4,22,23]. However, more recent field observations, which were subsequently confirmed experimentally, showed an increased pathogenicity of GI.2 strains, comparable to that of GI.1 strains in adult rabbits, with a similar incubation period and disease course [11,24,25]. Although several vaccines were commercialized for GI.1 [26,27] and/or GI.2 [25,28,29,30], and used to control the disease in farm and pet rabbits with very high efficacy in all cases [31], sometimes, in rabbitries, only the does are vaccinated. Thus, the disease, which is still endemic in several countries in wild rabbits and hare populations, can also affect unvaccinated domestic rabbits. 

Regarding GI.1 or GI.2 detection, several techniques of different complexity can be employed, such as immunohistochemistry, next-generation sequencing or RT-PCR (reviewed in [32]). Moreover, some commercial and non-commercial ELISA tests for which reagents are sold have been described for antigen detection [2,33,34]. Furthermore, a rapid test has been developed based on a lateral flow assay (LFA) to detect GI.2 but cannot discriminate between GI.1-positive samples and lagovirus-negative samples [35]. A combo rapid test, composed of two different strips, one for GI.1 detection and another for GI.2 detection, was developed by Certest Biotec SL (Zaragoza, Spain); however, it is not commercially available. Hence, in addition to the interest in a quicker and a cheaper antigen diagnostic test that can be used without the need for specific laboratorial equipment, new tests are needed for the differentiation of both variants. 

In this article, the development and evaluation of a new duplex LFA for the detection and differentiation of GI.1 and GI.2 in liver samples or liver exudates is described.

## 2. Materials and Methods

### 2.1. Sampling and Sample Preparation

A total of 96 liver homogenate samples were analysed, following the protocol described below. Of those 96 liver samples, 23 corresponded to GI.1-positive samples, 42 to GI.2-positive samples and 31 to lagovirus-negative samples. Furthermore, 40 liver exudates (liquid collected after liver freeze–thawing) were analysed: 24 corresponding to GI.2-positive samples and 16 to negative samples. These samples were kindly provided by the Research Centre in Biodiversity and Genetic Resources (CIBIO, Vairão, Portugal); French Agency for Food, Environmental and Occupational Health & Safety (ANSES, Ploufragan, France); Centro de Investigación en Sanidad Animal—Instituto Nacional de Investigación y Tecnología agraria y alimentaria (CSIC—INIA, Valdeolmos, Madrid, Spain); and 14 negative samples from animals collected in an authorized market. Samples were previously classified as negative or positive, GI.1 or GI.2, by the use of specific RT-PCR, of GI.2 real-time RT-PCR or the sequencing of lagovirus-specific RT-PCR products [3,4,5,7,17]. Furthermore, some liver samples were analysed by a commercial RHDV ELISA (INgezim^®^ RHDV DAS, Eurofins Ingenasa S.A., Madrid, Spain) to detect GI.1 and GI.2 antigen [21,36].

Sample preparation of the liver samples consisted of two steps: liver homogenization and homogenate extraction. First, 1 g of liver was weighed, cut, and vigorously squashed, followed by the addition of the same volume (1 mL) of phosphate-buffered saline (PBS) and continuous squashing to form a homogenate. Afterwards, using a pipette, 0.75 mL of homogenate was taken and placed in a tube containing 0.5 mL of PBS. The mixture was vigorously stirred for 30 s and centrifuged for 5 min at 1700× *g*. Finally, the supernatant was collected and kept at −80 °C for analysis.

Additionally, to decrease the complexity of the sample preparation in farms where lab equipment may be unavailable, a rapid extraction procedure was designed. A piece of liver was placed in an Eppendorf tube and the same amount of PBS, calculated by the naked eye, was added (*w*/*v*). Then, the tube was shaken for 30 s by hand, and the supernatant was collected using a plastic pipette.

### 2.2. Antigen Preparation (Virus Like Particles)

Virus-like particles (VLPs) were produced that expressed GI.1 VP60 and GI.2 VP60, as previously described [37,38]. Furthermore, Nodavirus VLPs (Nervous Necrosis Virus genotype SJ, SJNNV) were employed as negative control. In brief, Sf9 cells were infected with a recombinant baculovirus expressing the capsid protein of SJNNV. Three days after infection, cells were lysed with bicarbonate buffer and the protein contained in the supernatant was precipitated using 20% ammonium sulfate. After centrifugation, the pellet was suspended in PBS, purified using a 10–40% sucrose gradient and dialyzed in PBS.

### 2.3. Immunization of Mice, Development and Selection of Specific Monoclonal Antibodies

Four 10-week-old female BALB/c strain mice were subcutaneously injected with 50 µg of purified GI.2 VLPs in complete Freund’s adjuvant (CFA) (Difco, Franklin Lakes, NJ, USA). Further doses were performed with 50 µg of OVA-peptide at 2-week intervals to obtain peptide-specific antibodies. This peptide was based on the seven hypervariable regions (V1–V7) of the C-terminal region of the capsid protein of GI.2 VP60, constituting the protrusion (P) domain [39]. Mice received a final booster injection with 25 µg of purified GI.2 VLPs in PBS for 3 consecutive days prior to hybridoma fusion. 

To produce hybridomas-secreting, virus-specific MAbs, a modified version of the previously described procedure [40] was carried out. X63/Ag 8653 myeloma cells were fused with splenocytes from immunized mice using polyethylene glycol (PEG). After fusion, hybrid cells were eluted in Dulbecco′s Modified Eagle′s Medium (DMEM) containing 15% fetal calf serum (FCS) and coated on 96-well plates at a concentration of 0.5 × 10^6^ cells/well. After 24 h at 37 °C, 2-hypoxanthine-aminopterin-thymidine medium with hybridoma cloning factor was added and renewed every 48 h over a period of 12 days. At day 15, the culture medium of the hybridomas was screened for reactivity by ELISA with the corresponding GI.1 and GI.2 VLPs, as well as non-related Nodavirus VLPs. Antibody-producing hybridomas were cloned by limiting dilution at least four times. Antibody selection was performed using the corresponding VLPs and the unrelated Nodavirus VLPs (negative control). Finally, the selected hybridomas were grown using bioreactors and the resulting MAbs were purified from cell supernatant by affinity chromatography.

### 2.4. Characterization of Monoclonal Antibodies

The specificities of the MAbs were determined by indirect ELISA. The procedure was set up after titrating the antigen concentration and the conjugate dilution. In brief, the purified MAbs were incubated in 96-well microtiter plates coated with 0.2 µg/well of purified recombinant GI.1, GI.2 or Nodavirus VLPs and incubated overnight. The plates were blocked with Stabilzyme Select (SurModics, Inc., Eden Prairie, MN, USA) for 1 h at room temperature (RT). The bound antibody was detected with a 1/10.000 dilution of horseradish peroxidase (HRP)-labeled goat anti-mouse immunoglobulin (Sigma-Aldrich). After 1 h incubation at RT, followed by extensive washing, the plates were incubated with tetramethylbenzidine (TMB) substrate (Enhanced K-Blue TMB; Neogen Corporation, Lexington, KY, USA), and the reaction was stopped by the addition of 0.5 M sulfuric acid. The absorbance was measured at 450 nm in a Multiskan Ascent ELISA reader and the cut-off value was twice the negative control signal.

The isotypes of the MAbs were determined by ELISA, using specific anti-mouse subtype antisera (Sigma, Burlington, MA, USA).

Furthermore, immunofluorescence assay was performed by growing Sf9 cells on 8-chamber polystyrene vessels on tissue-culture-treated glass slides (Thermo Fisher Scientific, Waltham, MA USA). Cells were inoculated with BacGI.1 (expressing GI.1 VP60) and BacGI.2 (expressing GI.2 VP60). At 70 h post infection, the cells were washed with PBS and fixed with 95:5 acetone:methanol for 15 min at −20 °C. The cells were subsequently incubated with the primary antibody for 2 h at RT, and then with the secondary antibody for 1 h. After each incubation period, the cells were washed three times with PBS. Finally, the cells were embedded in glycerol-PBS (9:1 *v*/*v*) and images were taken using a Nikon Eclipse TS 100 microscope with a lamp at 470 nm and 46% intensity (CoolLED pF-100) for 100 ms. As negative control, cells were incubated with a non-related mouse monoclonal antibody and, furthermore, Sf9 insect cells were infected with a baculovirus expressing Nodavirus VLPs (data not shown).

### 2.5. Preparation of Lateral Flow Devices and Test Procedure

After analysing the battery of MAbs obtained, one of them was selected for use in the assay, named 16F12, which was specific to GI.2. Furthermore, a Mab (2E11) that recognized both genotypes and had been described previously [41] was also employed. The qualitative assay was a double duplex antibody sandwich LFA, in which the two antibodies were used as capture and/or detector reagent. Regarding the capture reagents that were dispensed in a nitrocellulose membrane, the test consisted of two test lines: one for the detection of both GI.1 and GI.2 and a second one for the detection of GI.2, using 2E11 and 16F12, respectively. Moreover, a control line containing a monoclonal antibody specific for biotin was dispensed. 

On the other hand, the detector reagent consisted of a 300-nm diameter red carboxyl-modified latex microspheres covalently coated with 2E11 MAb and 300-nm diameter blue nanoparticles coated with the control protein, both of which were dispensed onto the conjugate pad. Finally, the nitrocellulose membrane, conjugate pad and absorbent pad were assembled and then cut to strips that were placed in a plastic house.

The test procedure consisted of the addition of 20 µL of liver supernatant or liver exudate to the round window of the device, followed by 3 drops of running buffer. Results must be read 10 min after the addition of the buffer. The appearance of the two test lines (T1 and T2) or only T2 line indicates a positive result for GI.2, whereas the appearance of the T1 line indicates a positive result for GI.1. The appearance of the control line © is mandatory to consider the test valid.

## 3. Results

### 3.1. Characterization of Monoclonal Antibodies

A total of 10 MAbs were obtained from the fusion between myeloma cells and splenocytes from mice immunized with GI.2 VLPs and OVA-peptide, and named 11D10, 11C11, 11F3, 16F12, 14E11, 16H7, 12A1, 13A7, 13F4 and 16F9. 

An indirect ELISA, using the GI.1, GI.2 and non-related VLPs, was performed to test the specificity of each MAb. The results showed that the MAbs could be classified into two groups according to their specificity: five of them belonged to the GI.2-specific group (11D10, 11C11, 11F3, 16F12, 14E11) and the other five to the common GI.1 and GI.2 group (16H7, 12A1, 13A7, 13F4 and 16F9) (Table 1).

The isotypes of the MAbs were analyzed: antibody 16F12 was classified as IgG1; 11D10, 11F3, 14E11 and 12A1 were classified as IgG2a; 11C11, 13A7, 13F4 and 16F9 were classified as IgG2b; and 16H7 was classified as IgG3. 

To complement the results obtained by the indirect ELISA, the immunofluorescence assay was performed as described in the Materials and Methods section with the most promising MAbs. Three out of ten were selected, corroborating the previously obtained specificity (Table 1). An example of the immunofluorescence results is depicted in Figure 1.

### 3.2. Development of the Duplex Lateral Flow Assay

A new duplex lateral flow assay was developed for the detection and differentiation of GI.1 and GI.2. First, the MAbs that yielded the best signal and lowest unspecific background were selected: 2E11 and 16F12. Then, a set of assay geometries, reagent concentrations, extraction and running buffers, as well as nitrocellulose membranes, sample and conjugate pads, were evaluated. 

The assay consisted of the addition of 20 µL of the sample (extracted liver homogenate sample or liver exudate) to the device, followed by 3 drops of running buffer. After ten minutes, the visual observation of different coloured lines illustrates the presence or absence of GI.1 and GI.2, as shown in Figure 2. If only the blue line is observed (control line, C), the sample is negative for both lagovirus (1A). If C and T1 lines are visible, the sample is positive for GI.1 (1B). If the three lines (C, T1 and T2) are observed, the sample is positive for GI.2 (1C). If only T2 line is observed, the sample is positive for GI.2 but contains a low viral burden (1D).

### 3.3. Analytical Sensitivity

To determine the analytical sensitivity of the test, serial dilutions of the GI.1 VLPs and GI.2 VLPs were performed in running buffer and consequently analyzed. As shown in Figure 3, the limit of detection (LoD) of the duplex is lower for GI.2 VLPs than for GI.1 VLPs, detecting 0.08 µg/mL of GI.1 VLPs and 0.02 µg/mL of GI.2 VLPs.

### 3.4. Determination of Diagnostic Sensitivity and Specificity

With the aim of determining diagnostic sensitivity and specificity using field samples, liver homogenate and exudate samples were processed as described in the Materials and Methods section. Considering all the liver samples, the results indicated a specificity of 100% and a sensitivity of 93.8% compared with the RT-PCR results. When considering the group containing only GI.1-positive samples, sensitivity slightly changed to 93.1%. Regarding the group containing only GI.2-positive samples, sensitivity increased to 95.5%, while specificity was maintained in all cases (Table 2). These results were compared with the results obtained when analysing the samples by the commercial ELISA test and obtained an excellent agreement (weighted kappa of 0.977) (Figure 4).

The rapid extraction procedure described in the Materials and Methods was tested selecting by five positive and five negative liver samples. The results were the same as those obtained with the previously described laboratory procedure. Additionally, regarding liver exudates, only 24 GI.2-positive and 16 negative exudate samples could be analysed. Sensitivity among GI.2 samples was 95.8%, while specificity was 100% (Table 2). When considering the two panels of different types of samples together, the sensitivity was 94.4% (87.4–98.1, 95% CI), whereas specificity was maintained at 100% (92.4–100.0, 95% CI).

## 4. Discussion

Rabbit Haemorrhagic Disease is a deadly disease for rabbits that belongs to the OIE-list notifiable diseases. A rapid diagnosis of the disease can help in the prevention of its spread, especially in farms. Here, the development and evaluation of a rapid test for the detection and differentiation of GI.1, and the later reported genotype GI.2, in liver and liver exudate samples are described. 

First, GI.1 VLPs and GI.2 VLPs were serially diluted and analyzed to determine the analytical sensitivity, obtaining a lower LoD for GI.2 VLPs. This could indicate that the test is more sensitive with liver samples from GI.2-infected rabbits than with liver samples from GI.1-infected rabbits, but in the absence of viral quantification of samples, this remains a hypothesis. 

The high diagnostic sensitivity and specificity values achieved when evaluating liver samples after their homogenization and extraction indicated that the test could be applied as a valuable alternative to the reference technique, RT-PCR, when performing initial screenings. In addition, sensitivity was 93.1% and 95.5% for GI.1- and GI.2-positive samples, respectively, and these results are in accordance with the analytical sensitivity obtained with VLPs, since they are higher for GI.2 than for GI.1 samples. These samples were also analyzed using a commercial ELISA (INgezim^®^ RHDV DAS), which found only one discordant sample, corresponding to a sample classified as positive by ELISA but negative by RT-PCR and the described duplex LFA (Figure 4).

Employing liver exudates as a sample for virus detection showed the advantage of not having to process the sample, which would be useful for veterinarians when an outbreak in a farm is suspected. With the assessed sample panel, although GI.1-positive samples could not be analysed, the sensitivity with GI.2-positive samples was as high as 95.8%, very similar to the result obtained using liver homogenate samples, without compromising the specificity.

More results are needed to determine if the rapid extraction procedure (supernatant from a piece of liver shaken in PBS) could be applied to the liver samples, avoiding the complexity that this could involve in farms. The results obtained with five liver samples indicated that this would be feasible. Although liver is mainly used in viral diagnostic methods, since it contains the highest virus titre [2], the next step would be to adapt this assay for urine, faeces or respiratory secretions that are also shown to contain the virus [1], as well as with hare samples, due to their susceptibility to GI.2 infection. The other matrices could be very useful in disease surveillance in wild and farming rabbits, as the sample is simpler to extract in live and dead animals that could be encountered in the field.

## 5. Conclusions

These results show that the newly developed duplex lateral flow assay is a suitable tool for the differential diagnosis of GI.1 and GI.2 outbreaks in a single strip. It could be used as a reliable and rapid first diagnostic test for the surveillance of rabbit haemorrhagic disease, especially in farms and in the field. This rapid diagnosis should subsequently be confirmed by a competent national veterinary diagnostic laboratory.

## Figures and Tables

**Figure 1 biology-11-00401-f001:**
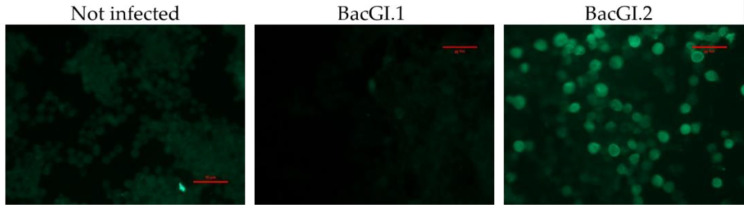
Detection of VP60 protein by the MAb 16F12 in non-infected and infected with BacGI.1 (expressing GI.1 VP60) or BacGI.2 (expressing GI.2 VP60) Sf9 insect cells by immunofluorescence. No signal was observed with the negative controls (data not shown). Scale ruler in red: 10 µm.

**Figure 2 biology-11-00401-f002:**
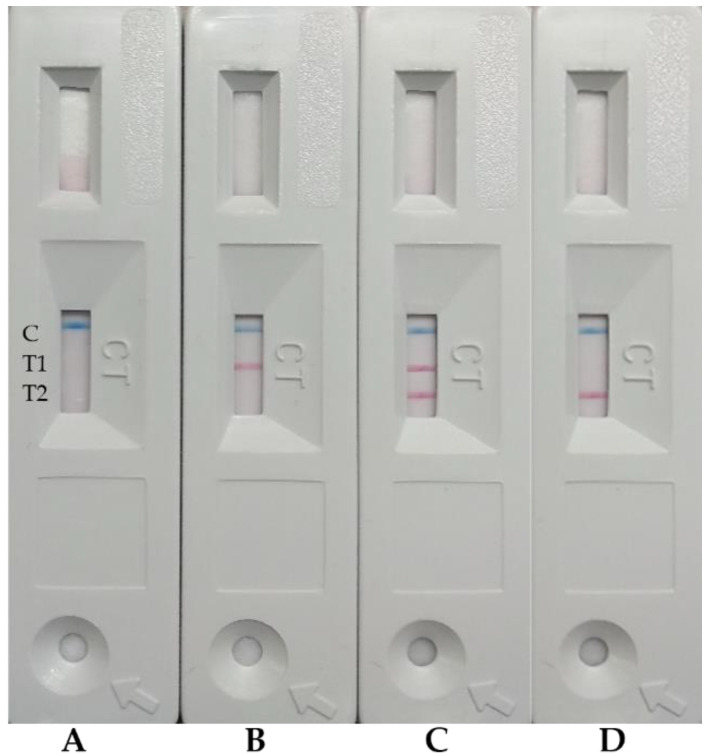
Visual interpretation of the GI.1 and GI.2 duplex LFA. (**A**) Sample negative for GI.1 and GI.2. (**B**) GI.1-positive sample. (**C**) GI.2-positive sample. (**D**) GI.2-positive sample containing low viral burden.

**Figure 3 biology-11-00401-f003:**
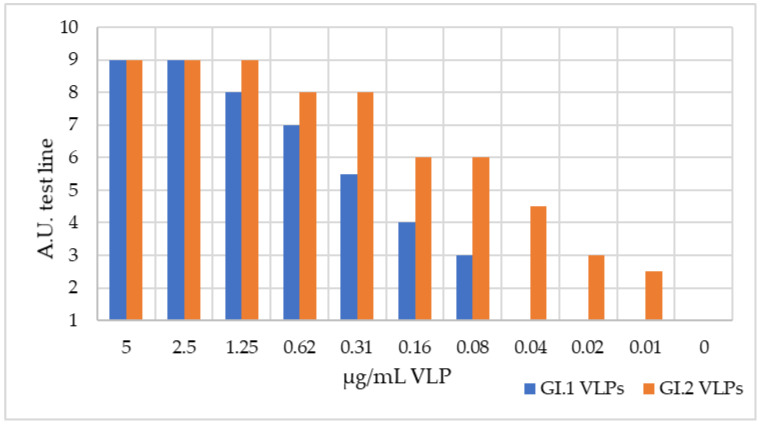
Analytical sensitivity of the duplex LFA. Serial dilutions of GI.1 VLPs and GI.2 VLPs and their corresponding result (A.U.).

**Figure 4 biology-11-00401-f004:**
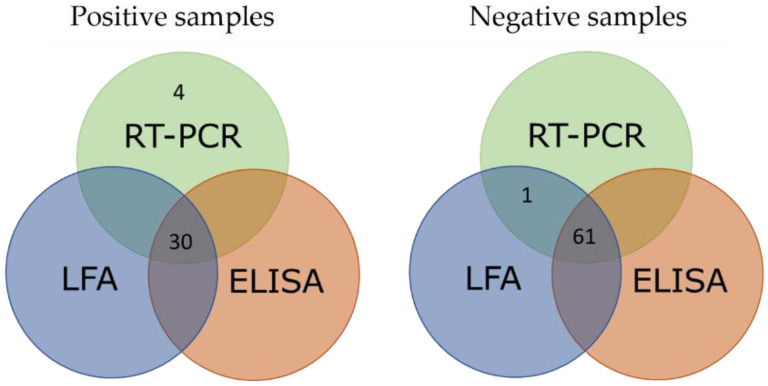
Venn diagram’s analysis of GI.1- or GI.2-positive and negative liver samples by RT-PCR, ELISA and LFA.

**Table 1 biology-11-00401-t001:** Characterization of monoclonal antibodies.

Specificity	MAb	Isotype	IF Result
GI.2	11D10	IgG2a	GI.2 VLPs
	11C11	IgG2b	ND
	11F3	IgG2a	ND
	16F12	IgG1	GI.2 VLPS
	14E11	IgG2a	ND
Common to GI.1 and GI.2	16H7	IgG3	GI.1 and GI.2 VLPs
	12A1	IgG2a	ND
	13A7	IgG2b	ND
	13F4	IgG2b	ND
	16F9	IgG2b	ND

**Table 2 biology-11-00401-t002:** Diagnostic performance of the developed assay compared to RT-PCR.

Detection	Liver Samples		Liver Exudates	
	Sensitivity % (95% CI)	Specificity % (95% CI)	Sensitivity % (95% CI)	Specificity % (95% CI)
GI.1	93.1 (71.9–98.7)	100 (88.7–100.0)	ND	100 (79.2–100.0)
GI.2	95.5 (84.5–99.3)	100 (88.7–100.0)	95.8 (78.8–99.3)	100 (79.2–100.0)

## Data Availability

Data are contained within the article.

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
