# Peer review of "Development and Evaluation of a Duplex Lateral Flow Assay for the Detection and Differentiation between Rabbit Haemorrhagic Disease Virus Lagovirus europaeus/GI.1 and /GI.2"

_biology, 2022, doi:10.3390/biology11030401_

Round 1
Reviewer 1 Report
AUTHORS
This is a study describing the development and testing of a duplex lateral flow assay for antigen detection that allows simultaneous detection and differentiation of RHDV genotypes GI.1 and GI.2. For this authors have used a group of GI.1‐ or GI.2‐infected and non‐infected rabbit liver samples and liver exudates (136 samples) and have obtained a total sensitivity of 94.4 % and specificity of 100%. The value of such protocol and data is high and warrants visibility. The manuscript is also written correctly, making use of good English language. For the above I advise publication after minor modifications are made
Authors use for several times the word “cohort” when they should be using the word “group” or “groups” and there is no follow-up
How did authors define the concentration of coating VLPs on the ELISA plates? Why 0.2 μg/well? Provide the time for coating (overnight?)
Again the same question for the 1/10.000 dilution of horseradish peroxidase. Why the concentration and time of incubation
What was the cutoff value for the ELISA?
Please describe the usage of positive and negative controls for the immunofluorescence assay
Reviewer 2 Report
Comments to the Author
The paper by Fresco-Taboada et al. describes the development and evaluation of a duplex lateral flow assay for the detection and differentiation between rabbit haemorrhagic disease lagovirus europaeus/GI.1 and GI.2.
The pen side test is to be used in farms, and in the field, allowing the rapid “laboratorial” diagnosis, for immediate implementation of control measures.
General observations
I was surprised by the fact that the authors did not mentioned in any part of their manuscript that there is already a similar product (pen side test) available in the market for the detection of GI.1 and GI.2 (CerTEST Biotec). This should be added to lines 77-80. Since the first author is staff of INGENAZA, and INGENAZA and CerTEST Biotec are competitor companies, the absence of this relevant information could suggest a conflict of interests.
I fully agree with the authors that the rapid diagnosis “on the spot” is useful to take measures. However, I also believe that it is important to point out to the readers that this rapid diagnosis must be subsequent confirmed by more sensitive and specific methodologies at the competent national laboratories.
Regarding the utility of the differential diagnosis, after the emergence of GI.2 in 2010, the GI.1-strains have become a minority of the strains circulating in Europe, due to the rapid replacement of GI.1 by GI.2 strains, that will eventually lead to the complete GI.1 disappearance.
In fact, in Europe, GI.1-strains are only very occasional reported (Abrantes et al. Genes. 2020; Szillat et al., 2020)).
However, in China, where GI.2 only emerged recently (in 2020) in the Sichuan province, the device may prove useful.
The manuscript is well written and the experimental design is clear.
In particular:
Line 16. Hepatitis, splenomegaly and haemorrhages are not symptoms. Please correct.
Line 27. “ and replaced” or “and replacing”
Line 41. “Since than”
Line 289. Please indicate the RT-PCR method.
Line 303. “as those” instead of “as the ones”.
Discussion section
Line 347-349. The last sentence also applies to the CerTEST device. Please add the unique specificities of the device described (differential diagnosis).
